# Research on Construction Land Use Benefit and the Coupling Coordination Relationship Based on a Three-Dimensional Frame Model—A Case Study in the Lanzhou-Xining Urban Agglomeration

**Weiping Zhang [1], Peiji Shi [1,2,\*] and Huali Tong [1]**

1 College of Geography and Environmental Science, Northwest Normal University, Lanzhou 730070, China; 2021120199@nwnu.edu.cn (W.Z.); tonghl@nwnu.edu.cn (H.T.)
2 Gansu Engineering Research Center of Land Utilization and Comprehension Consolidation, Northwest Normal University, Lanzhou 730070, China
\* Correspondence: shipj@nwnu.edu.cn; Tel.: +86-138-9366-5158

**Abstract:** Coordinating the social, economic, and eco-environmental benefits of construction land use has become the key to the high-quality development of Lanzhou-Xining urban agglomerations (LXUA). Therefore, based on the coupling coordination connotation and interaction mechanism of construction land use benefit (CLUB), we measured the CLUB level and the coupling coordination degree (CCD) between its principal elements in LXUA from 2005 to 2018. Results showed that: (1) The construction land development intensity (CLDI) in the LXUA is generally low, and spatially presents a dual-core structure with Lanzhou and Xining urban areas as the core. (2) The comprehensive construction land use benefit has increased over time, but the overall level is not high. The spatial differentiation is obvious, and the core cities (Lanzhou and Xining) are significantly higher than other cities. (3) The regional differences in the subsystem benefit of construction land use are obvious. The social benefit and economic benefit showed a "convex" shape distribution pattern of "high in the middle and low in the east and west wings", and regional differences of economic benefit vary greatly. The eco-environmental benefit was relatively high, showed a "concave" shape evolution in the east–west direction. (4) In addition, the CCD of the CLUB were still at a medium–low level. The higher the administrative level of the city, the better the economic foundation, and the higher or better the CCD of the social, economic, and eco-environmental benefits. (5) The CCD is inseparable from the influence of the three benefits of construction land use. Therefore, different regions should form their own targeted development paths to promote the coordinated and orderly development of LXUA.

**Keywords:** construction land development intensity; construction land use benefit; coupling and coordination relationship; spatiotemporal evolution; Lanzhou-Xining urban agglomeration

## 1. Introduction

Land is the basic carrier of all social and economic activities, and can provide space and resources for human production and life [1]. As an important part of land resources, construction land mainly includes urban construction land, rural residential land, transportation land, water conservancy land, and other construction land. Its utilization is not only related to the development of the secondary and tertiary industries of the national economy, but also has an important impact on the coordinated development of urban and rural areas [2,3]. Land use benefit refers to the comprehensive output of social, economic, and eco-environmental benefits obtained by human capital, labor, and technical input, which is related to the sustainable development of a region [4].

At present, due to rapid global urbanization, most developed countries and regions are striving to complete the simultaneous optimization of land use in society, economy, and

ecology, and have made some achievements [5–7]. As the main target of urban expansion in the coming decades [8], driven by industrialization, new urbanization and modernization, the urban population of developing counties has increased rapidly, and the scale of urban construction land development has been expanding continuously. Especially in China, the urbanization rate has increased from 17.90% in 1978 to 59.58% in 2018 [9]. However, with the rapid increase in urbanization level, the contradiction between man and land has become increasingly acute, and problems such as extensive land development and utilization, disorderly expansion, and land pollution have frequently occurred, which has led to an imbalance between social, economic, and eco-environmental benefits of land use, especially in urban agglomeration areas where social and economic activities are more prevalent [10,11]. In addition, urban sprawl will inevitably affect the implementation of cultivated land protection policies and even national food security [12]. Therefore, it is of great significance to study the development intensity and multi-dimensional use benefits of construction land for the rational and optimal allocation of land resources and the sustainable development of urban and rural areas.

Scholars have done a lot of research on land use benefit, mainly focusing on the evaluation of land use benefits, the diversification and applicability of research methods and models, and the excavation of influence mechanisms. In the early days, people only paid attention to the economic benefits of land. For example, the law of land rent in western economics laid a theoretical foundation for the study of the economic benefits of land use [13]. Fulton et al. also verified that urban land expansion was closely related to urban land economic benefits [14]. However, urban sprawl also leads to the increase of social and ecological costs such as prolonged commute time, waste of resources, and destruction of ecological environment pollution [15]. Therefore, while discussing the principle of maximum return brought by non-renewability and restriction of land in principles of land economics, scholars emphasize that land use should meet social goals such as wealth production, fair distribution, and ecological protection; and improve the overall benefit of land use by means of political, legal and economic leverage [16]. With the challenge of sustainable development, people are more aware that the goal of land use is the sustainable use of land resources, which must take into consideration the coordinated development of social, economic and eco-environmental benefits [17,18]. According to previous research, the research object and perspective of land use benefit is not only to evaluate the land use benefit, but also to change the coordination relationship between land use benefit and urbanization and other factors. For example, Zhang et al. analyzed the coupling and coordination between urban land use efficiency and urbanization in 34 prefecture-level cities in the three northeastern provinces, and found that although there is a mutual response relationship between them, urbanization pays too much attention to development speed and despises development quality, resulting in low overall development level and low coupling and coordination degree [19]. He et al. constructed the theoretical framework of land use benefit and industrial structure evolution, and found that industrial structure evolution has an obvious single effect on land use benefit [20].

At the same time, there are regional differences in the study of land use benefits. In developed regions such as America and Europe studies of combined land use, urban expansion, and ecological environment management to try to analyze and evaluate ecosystem services and economic benefits through land use modeling, land protection, and planning behavior [21–23]. While in developing countries, the research is more related to urbanization and urban sprawl, focusing on the rational use of urban land, coordination of urban and rural land, and management strategies for the process of rapid urbanization [24,25]. Research methods and models of land use benefits are also hot topics that scholars pay attention to. The existing research is mainly based on the construction of a land use benefits index system, with the data envelopment analysis model, neural network model, multi-objective linear programming model, SWAT model, coupling degree model, and other methods [26–30]. With the development of science and technology, GIS, remote sensing and spatial measurement methods have been introduced into the study of land use and

land expansion, and the spatial monitoring and analysis functions of these methods have been applied to realize the visual expression of the research results [31,32]. Of course, due to data availability and the research objects being different, the weighting methods and indicators selected are also different. Methods such as the analytic hierarchy process (AHP), coefficient of variation method, expert scoring method, and entropy weight method are widely used [33–35]. Based on this, Ran et al. used the Friedman chi-square test, Spearman correlation, consistency test, and one-way analysis of variance to compare different subjective and objective weighting methods, they found that the comprehensive index method, rank-sum ratio method, entropy method, and the integrated entropy methods all have significant statistical characteristics, and each has its own advantages and disadvantages. A more scientific, comprehensive approach is required [36]. Research on the influencing factors of land use benefits generally focuses on economic, social, transportation, and political factors [37–40]. Some studies have also found that the determinants are related to urbanization and industrialization, accessibility, and economic transformation [41,42]. These studies provide policy and guidance for improving land use efficiency and promoting sustainable development of cities.

From the current research progress, we also find that most of the land use benefits studies focus on the desirable outputs of limited land use, such as economic benefit, while there are few studies on undesirable outputs such as pollution and industrial emissions from land use. In addition, the study areas are mainly concentrated in areas with a high economic level, while the research on land use benefits of the northwest region China, where the ecological environment is relatively fragile, is relatively weak. Therefore, this paper selects the LXUA which is the important urban agglomeration on the Silk Road Economic Belt as an example. On the basis of elaborating and analyzing the connotation and mechanism of the coupling coordination of multi-functional benefits of construction land use, a three-dimensional framework system of CLUB is constructed including society, economy, and eco-environment. Secondly, using an entropy method, a composite index model and coupling model, and the application of CLUB is demonstrated. Finally, according to the evolution law and spatial differentiation characteristics of the coupling coordination relationship of CLUB in different regions, an optimization strategy for sustainable land use development is proposed according to local conditions. The overall workflow of the study is shown in Figure 1.

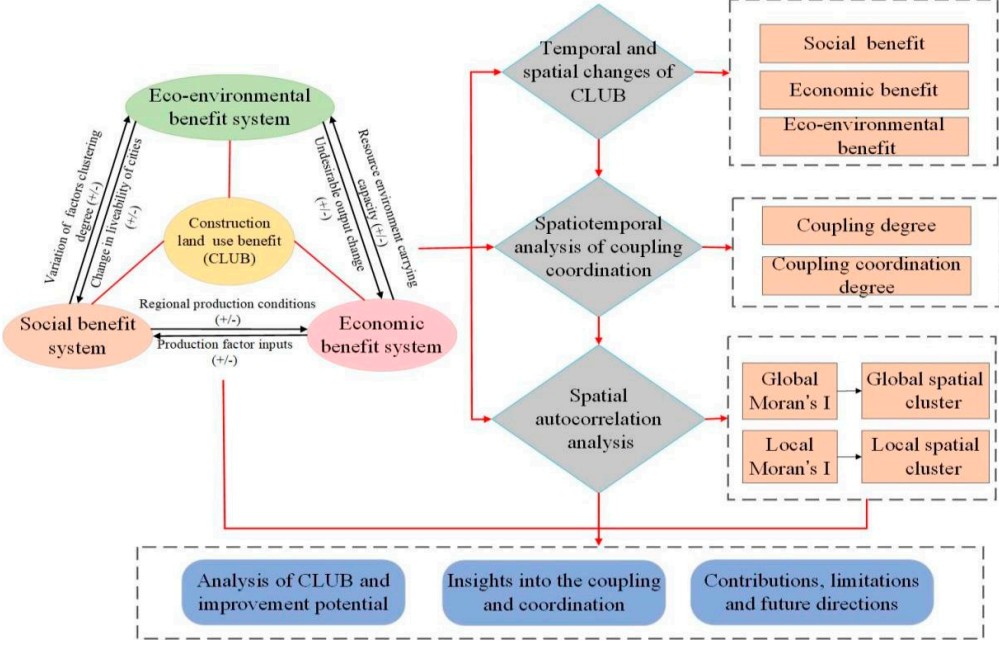

**Figure 1.** The coupling interaction relationship of the CLUB system and workflow of the study.

The structure of this article is as follows. Section 2 explains the conceptual framework of the interactive coupling relationship of the CLUB system. Section 3 introduces the current situation of LXUA, data sources, and the research method. Section 4 analyzes the spatiotemporal differentiation of the CLUB based on the CLDI, and explores the coupling and coordination relationship between the social, economic, and eco-environmental benefits of the construction land use. Section 5 discusses and analyzes the important conclusions. Section 6 summarizes the main conclusions and future research directions.

## 2. Interaction Coupling Mechanism of CLUB

The economic, social, and eco-environmental benefits of construction land use are closely related, restricted, and promoted by each other [43]. On one hand, land is an important carrier of all human activities. With the rapid advancement of urbanization, construction land has been continuously developed and utilized, providing space and resources for various activities. Its quantity and quality are closely related to the social and economic benefits. If people develop land and do not exceed the resource's environmental carrying capacity, we can get economic benefit continuously. If the economic benefit is invested in improving people's livelihood and infrastructure construction, good social benefit will be produced. The improvement of social benefit will improve regional production conditions, promote further development of the regional eco-environment. On the other hand, in the process of land use, due to the limitation of technology, capital, and knowledge level, people will have negative economic scale effects, which can lead to environmental pollution, ecological systems destruction, and affect the eco-environmental benefit of land. The destruction of the eco-environment will worsen the local environmental conditions and affect the social benefit of land use. Moreover, it will also reduce the local resources environmental carrying capacity, resulting in the loss of land economic benefit. To sum up, only when the three achieve dynamic coordination and balance, can they promote the effective improvement of the whole system benefits to a greater extent and maximize the benefits (Figure 1).

## 3. Materials and Methods

### 3.1. Study Area

The LXUA is a $9.75 \times 10^4$ km regional urban agglomeration in the upper reaches of the Yellow River in the inland northwest China (Figure 2), including nine cities, Lanzhou, Baiyin, Dingxi, Linxia, Xining, Haidong, Hainan, Haibei, and Huangnan, and a total of 39 counties. The population concentration degree is relatively high, and it is an important radiation center and growth pole in Northwest China. As of 2018, the GDP of the LXUA reached CNY 515.59 billion, accounting for 51.13% of the total of Gansu and Qinghai province. As the hub of the Silk Road Economic Belt and the Yangtze River Economic Belt, LXUA's geographical advantages have become increasingly prominent, especially the construction of the new land–sea passage in the west, which makes LXUA's hub position of the "Sixth Ring in the Middle" more prominent. This region has good resource endowment and belongs to a region with good soil–water combination conditions in Northwest China. However, the economic level within the urban agglomeration varies greatly, and the imbalance of regional development is outstanding. Therefore, under the background of the new round of western development, how to balance the benefit relationship between land use, promote the coordination between development, utilization and protection of land, and realize the promotion of comprehensive benefit of land use in the region, has become the primary task of high-quality development of urban agglomerations.

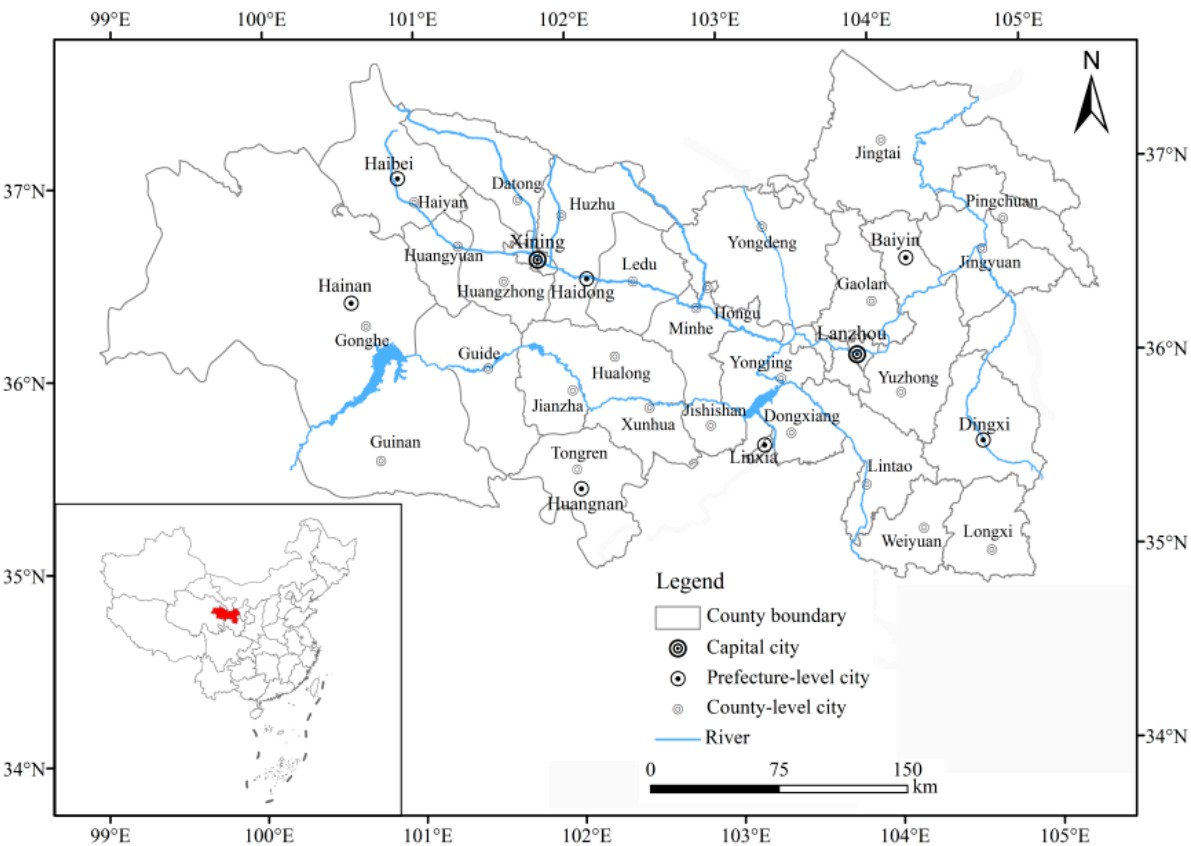

**Figure 2.** Study area.

*3.2. Data Sources*

This research takes 39 counties in LXUA as the basic research unit. The vector boundaries of urban agglomerations and the construction land data are all from the Resource and Environmental Science Data Center of the Chinese Academy of Sciences (http://www.resdc.cn accessed on 15 November 2021). According to the national county-level land use classification system, the data of the corresponding year was revised to extract and divide the construction land area data of each county. The socioeconomic data of each research unit was mainly derived from the Statistical Yearbooks of the Chinese County, Statistical Yearbooks of Gansu and Qinghai Province. The eco-environmental data was mainly derived from the environmental protection bureaus of cities and prefectures and field survey.

*3.3. Methods*

3.3.1. CLDI Measurement

Development intensity can be used to measure the extent of land development and utilization by human activities in a certain region. As for the development intensity, 20% is generally regarded as the standard line of livability and 30% as the warning line, globally. The National Land Planning Outline (2016–2030) proposes that the development intensity of China's land should not exceed 4.62%. This paper selects the proportion of construction land area to the regional total area proposed by Fan in the main functional zone as a measure of CLDI [44]. The equation is as follows:

$$\text{CLDI} = \frac{CLA}{TA} \times 100\% \tag{1}$$

In Equation (1), CLDI represents construction land development intensity, *CLA* is construction land area, and *TA* is the total area of region.

### 3.3.2. CLUB Index System

The CLUB mainly refers to the comprehensive output of social, economic, and eco-environmental benefits caused by the labor, capital, and technology invested by human beings in construction land. Therefore, drawing on previous studies [43,45,46], and combined with the connotation of CLUB and the scientific principle of index selection, and the availability of data, this paper constructed a three-dimensional evaluation index system including social, economic, and eco-environmental benefits (Table 1).

**Table 1.** Three-dimensional framework evaluation index system of CLUB.

| Criterion Layer | Indicator Layer | Attribute | Weight |
|---|---|---|---|
| Social benefits | Urbanization level (%) | + | 0.0534 |
| | Population density per unit construction land area (person/km$^2$) | + | 0.0646 |
| | Resident per capita net income (RMB/person) | + | 0.0511 |
| | Employed persons per land (person 10,000/km$^2$) | + | 0.0469 |
| | Per capita road area (m$^2$/person) | + | 0.0812 |
| Economic benefits | GDP per unit area (RMB 10,000/km$^2$) | + | 0.2377 |
| | Fiscal revenue per unit of construction land area (RMB 10,000/km$^2$) | + | 0.0683 |
| | GDP per unit construction land area of secondary and tertiary industries (RMB 10,000/km$^2$) | + | 0.0625 |
| | Retail sales of consumer goods per unit of construction land area (RMB 10,000/km$^2$) | + | 0.0769 |
| | Investment in fixed assets per unit construction land area (RMB 10,000/km$^2$) | + | 0.0593 |
| Eco-environmental benefits | Green space coverage in built-up areas (%) | + | 0.0373 |
| | Public green space per capita (m$^2$/person) | + | 0.0506 |
| | Discharge of industrial wastewater per unit construction land area (T/km$^2$) | - | 0.0368 |
| | Industrial waste gas emissions per unit construction land area (T/km$^2$) | - | 0.0335 |
| | Discharge of industrial solid waste per unit construction land area (T/km$^2$) | - | 0.0399 |

### 3.3.3. CLUB Index Weight Setting and Score Calculation

Data Preprocessing

Different indicators have positive and negative effects, in order to ensure the rationality of the evaluation results, it was necessary to standardize the data to ensure the uniformity of the dimensions of each indicator by the following:

$$x'_{ij} = \begin{cases} \frac{x_{ij} - x_{\min}}{x_{\max} - x_{\min}} & \text{Positive indicator} \\ \frac{x_{\max} - x_{ij}}{x_{\max} - x_{\min}} & \text{Negative indicator} \end{cases} \tag{2}$$

In Equation (2), $x'_{ij}$ represents the standardized value, $x_{ij}$ is the original data value of the *j*-th indicator, $x_{\max}$ and $x_{\min}$ are the maximum and minimum values of the *j*-th indicator.

Weight Calculation

Due to the different degrees of dispersion of different indicators, the impact on the comprehensive evaluation is also different. Therefore, this study used the more objective entropy method to calculate the index weight [46,47]. The weight results are shown in Table 1.

Comprehensive Value and Subsystem Score Calculation

According to the above standardized values and weights, the social benefit value ($U_{soc}$), economic benefit value ($U_{eco}$), and eco-environmental benefit value ($U_{env}$) of construction land use were calculated in combination with the composite index method [46]. Then, the comprehensive value ($U_{com}$) of construction land use of the *i*-th sample was obtained by using the weighted evaluation method of the criterion layer.

$$U_{soc/eco/env} = \sum_{j=1}^{n} \left( w_j \times x'_{ij} \right) \tag{3}$$

$$U_{com} = \alpha U_{soc} + \beta U_{eco} + \gamma U_{env} \tag{4}$$

In Equations (3) and (4), $U_{soc}$, $U_{eco}$, and $U_{env}$ represent the social benefit, economic benefit, and eco-environmental benefit of construction land use, respectively; $x'_{ij}$ is the standard value of the $j$-th index in three systems; $w_j$ represents the weight of the corresponding index. $U_{com}$ represents the comprehensive benefit of construction land use. $\alpha$, $\beta$, and $\gamma$ are undetermined coefficients, which represent the importance of the social, economic, and eco-environmental benefits of construction land use, respectively. Because the three need to coordinate and advance together and are almost equally important, this paper takes $\alpha = \beta = \gamma = 1/3$.

3.3.4. Coupling Degree (CD) and Coupling Coordination Degree (CCD)

Coupling is usually used to describe the degree to which two or more systems interact and affect each other [48,49]. This paper constructs a CD model among the three-dimensional benefits of social, economic, and eco-environmental. Based on the CD, this paper further draws on the CCD to measure the degree of harmony and consistency of the social, economic, and eco-environmental benefits of construction land use in the process of change [50]. The equations are as follows:

$$C = \left[ \frac{U_{eco} \times U_{soc} \times U_{env}}{\left( \frac{U_{eco} + U_{soc} + U_{env}}{3} \right)^3} \right]^{\frac{1}{3}} \tag{5}$$

$$D = \sqrt{C \times U_{com}} \tag{6}$$

In Equations (5) and (6), $C$ denotes the coupling degree of social, economic, and eco-environmental benefits of construction land use, and $C \in [0, 1]$. $D$ denotes the coupling coordination degree between social, economic, and eco-environmental benefits of construction land use, and $D \in [0, 1]$. The larger the $D$, the higher the CCD. When $C = 0$, the CD is extremely low, and the systems, or between elements within the system, are in a disordered state. When $C = 1$, the CD is the largest, and benign coordinated coupling is achieved between systems or between internal elements of the system.

On the basis of existing research, we found that many scholars have used the critical value method to classify CD and CCD. However, this paper does not subjectively carry out quantitative division, but chooses the objective quartile method. The CD and CCD are divided into four stages from low to high level: lower level, medium level, higher level, and benign (optimal) level.

**4. Results**

*4.1. Spatiotemporal Change of CLDI*

According to Equation (1), it was found that the CLDI in LXUA had an upward trend from 2005 to 2018. The average of CLDI in 2005, 2010, and 2018 were 6.53%, 7.31% and 8.54%, respectively. In 2005, CLDI was dominated by low-value and lower-value counties, accounting for 38.46% and 23.08%, respectively. In 2010, the CLDI showed a steady increase, compared with 2005, the proportion of low-value areas decreased, and the number of medium and high-value areas increased. By 2018, the CLDI showed an accelerating trend, the number of counties with a development intensity of 2% and above continued to increase. However, compared with cities in eastern and central China [31], the CLDI in LXUA is relatively low.

As shown in Figure 3, the CLDI differs on different spatial scales. On the provincial scale, the CLDI in Gansu area (8.63%) is higher than that in the Qinghai area (6.95%). On the prefecture-level city scale, Xining, Lanzhou, and Baiyin ranked the top three in terms of development intensity. At the county level, the CLDI in the municipal districts of Xining, Linxia, Lanzhou, and Baiyin all exceeded the national land development intensity of 4.60% in the National Land Planning Outline (2016–2030). In terms of changes in the entire urban agglomeration, the CLDI in the LXUA generally presents a "core-periphery" spatial structure with the Lanzhou and Xining urban areas as the core, and other peripheral areas

decreasing in turn. The closer it is to the city center, the stronger the gathering capacity of construction land and the larger the development scale. On the contrary, the further from the city edge, the CLDI gradually decreases.

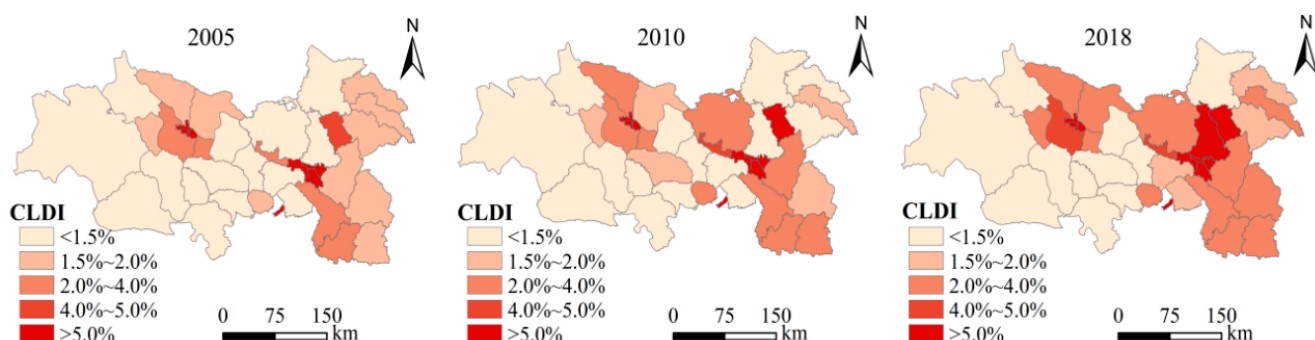

**Figure 3.** The spatial pattern of CLDI in LXUA.

### 4.2. Spatiotemporal Change of CLUB

As shown in Figure 4a, the social benefit of construction land use showed a trend of "first rise and then decline", and first rose from 0.28 in 2005 to 0.35 in 2010, and then declined to 0.33 in 2018. From Figure 4b, the variation coefficient of social benefit fell from 0.57 to 0.44, a decrease of 22.81%, which indicates that the regional differences of social benefit tend to narrow. From Figure 5a, the social benefit is characterized by a "convex" shaped spatial distribution. The social benefit of Lanzhou and Xining was always higher than that of other regions. Among them, the middle and high value regions are gradually concentrated in the central area connected with Lanzhou and Xining, while the social benefit in the peripheral counties of the urban agglomeration is low.

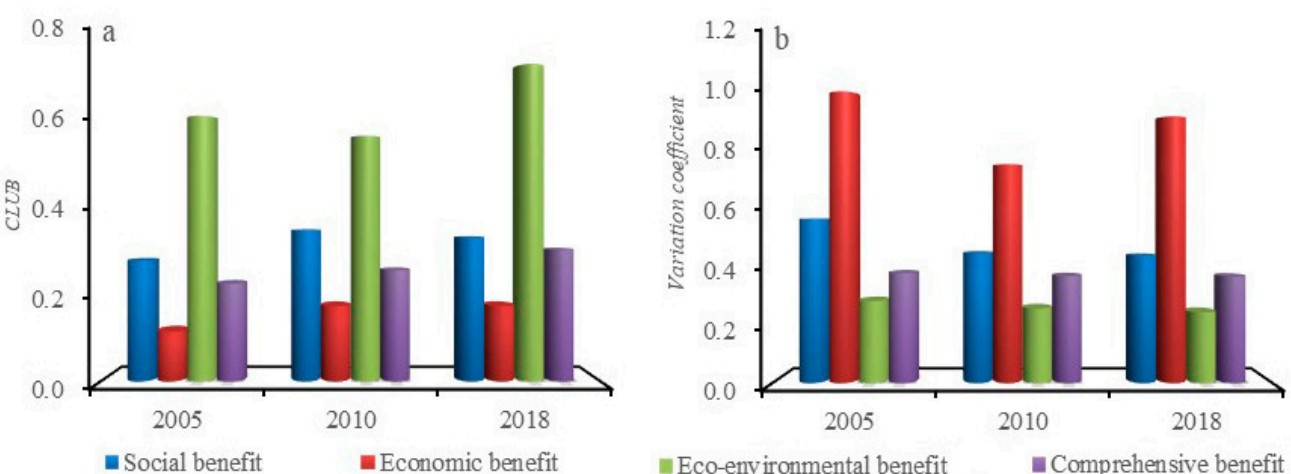

**Figure 4.** CLUB value and its variation coefficient. (**a**) Construction land use benefit level, (**b**) Variation coefficient of construction land use benefit.

As shown in Figure 4a, the economic benefit of construction land use from 2005 to 2018 showed a "continuous increase", and increased from 0.12 to 0.17, an increase of 41.67%. However, as can be seen in Figure 4b, the variation coefficient of economic benefit decreased from 1.01 to 0.93, with a small reduction of only 7.92%, which means that the regional differences of economic benefit were still large. From Figure 5b, the economic benefit presented a spatial pattern of "high in the west and low in the east, high in the middle and low on the outside". The middle and high value areas were concentrated in Huangzhong, Chengguan, and Ledu, and the low value areas were mainly distributed in the eastern fringe counties of the urban agglomeration.

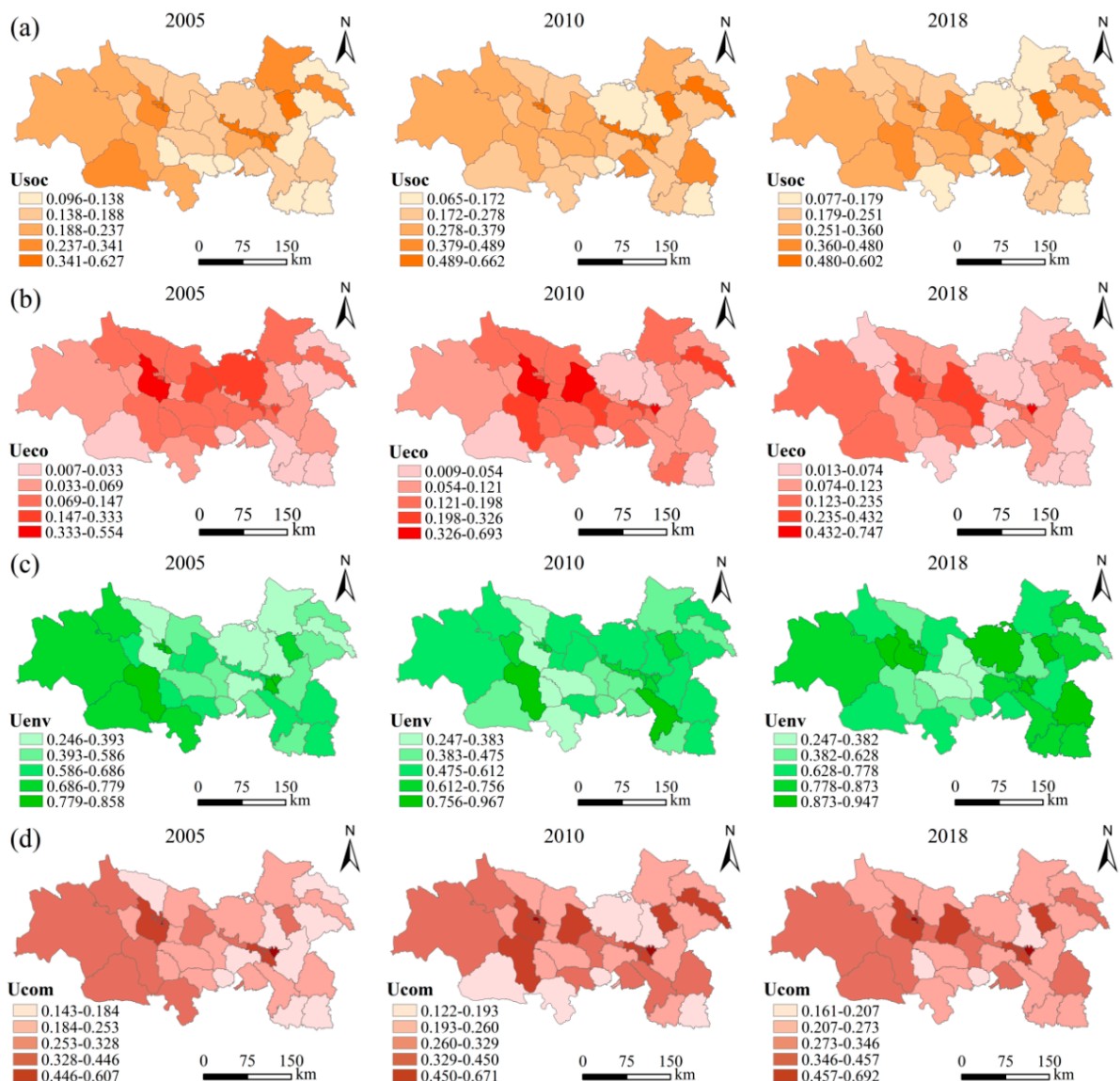

**Figure 5.** Spatial pattern of the CLUB in LXUA. (**a**) Social benefit, (**b**) Economic benefit, (**c**) Eco-environmental benefit, (**d**) Comprehensive benefit.

From Figure 4a, the eco-environmental benefit of construction land use showed a characteristic of "first decline and then increase". Itfirst declined from 0.61 in 2005 to 0.57 in 2010, and then increased to 0.73 in 2018. From Figure 4b, the regional differences of the eco-environmental benefit were smaller and tended to shrink slightly, with a shrinking rate of 10.71%. As can be seen in Figure 5c, the eco-environmental benefits were characterized by a "concave" shaped spatial distribution. In 2005, the high value areas were mainly concentrated in the western region. By 2018, the eco-environmental benefit of the eastern of the urban agglomeration had risen significantly, and showed a trend of "retreat from west to east".

As shown in Figure 4a, the comprehensive benefit was a continuous upward trend during the study period, and increased from 0.27 to 0.32, an increase of 18.52%. However, we can see from Figure 4b the variation coefficient of comprehensive benefit decreased from 0.38 to 0.37, with a decrease of only 2.63%, which indicates that the regional differences of comprehensive benefit demonstrated only a slight shrinking trend. From Figure 5d, the comprehensive benefit showed a distribution of "high in the middle and low in the outside", and the urban land use benefit with Lanzhou and Xining as the core were always higher than that of other counties. At the same time, the number of low-level counties

decreased significantly, from 20.52% to 7.69%. Overall, the CLUB of LXUA are mainly at low and lower level, and showed a sequential shift from low level area to lower level area.

### 4.3. Coupling and Coordination Analysis of CLUB

During the important period of socio-economic transformation and development of LXUA from 2005 to 2018, its land development, utilization structure, and output benefits were affected by population agglomeration, industrial development, and urban policies. As shown in Figure 6a, the coupling degree (CD) value C of CLUB generally fluctuated and rose, and the western counties were higher than the eastern counties. Higher and benign level counties of the CD gradually shifted from a scattered to centralized distribution, mainly in the western and central areas of the urban agglomeration. As the "back garden" of the central city, the demand for industrial development, infrastructure, and residential space in such areas may be in a stage of continuous increase, coupled with the strong promotion of the policy of linking the increase and decrease of construction land, which made the social, economic, and comprehensive benefits of construction land at a good level. Therefore, the CD is also high. Lower level counties of the CD were concentrated in the eastern and northern fringes of LXUA. Such areas were mostly development areas of traditional industry with slow economic development and a single structure, with primary industry accounting for a considerable proportion. Its low level of urbanization and industrialization, limited population agglomeration capacity, and slow expansion of built-up areas, resulted in social, economic, and comprehensive benefits of construction land use being low. Therefore, the CD is also lower.

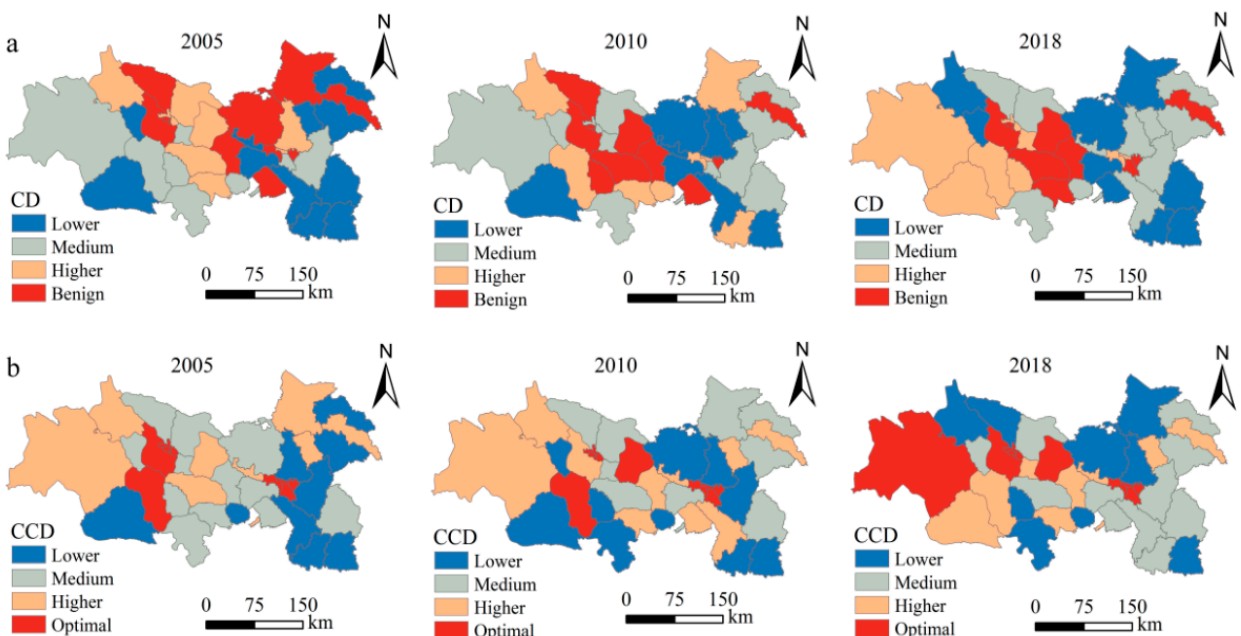

**Figure 6.** Spatial differentiation of the CD and CCD of social, economic, and eco-environmental benefits of construction land use in LXUA. (**a**) Coupling degree, (**b**) Coupling coordination degree.

To judge whether the input and output of regional construction land development is reasonable and orderly, in addition to considering its interaction and correlation, it is also necessary to explore the coupling coordination degree (CCD) of CLUB. Figure 6b showed that the spatial pattern of the CCD was basically consistent with the CD, but the CCD was generally lower than the CD. In Yongdeng, Gaolan, Jingtai, Yuzhong, Anding, Longxi, Haiyan, and Huangyuan counties, where the CD was at a medium-low level, their coupling characteristics showed a low-level orderly coordination state, and the CLUB in such areas was at a lower level of coordinated evolution, and the industrial development was mostly in the primary stage, so the response sensitivity to the improvement of construction land

use efficiency was slow. Correspondingly, the CCD of higher CD regions of construction land use benefit was also higher, which reflects a high-level synchronous evolution state. From the spatiotemporal evolution, the CCD showed a continuous upward trend, and a regional distribution with optimal level coordination stages did not change significantly from 2005 to 2018, and was mainly concentrated in Lanzhou and Xining urban areas. The medium and lower level counties accounted for 41.03% in 2018. Overall, the CCD between the social, economic, and eco-environmental benefits of construction land use in LXUA is still in the lower level coordination stage, and the CCD needs to be improved urgently.

### 4.4. Spatial Autocorrelation Analysis

Spatial autocorrelation can well express the spatial relationship of CCD between the CLUB, so as to further reveal the spatial connections and differences among the research units [50]. Figure 7 shows the Moran's I of the CCD from 2005 to 2018. The Moran's I values are all positive, and $p < 0.01$, which indicates that the CCD had a significant positive spatial autocorrelation. The Moran's I showed a fluctuating trend. From 2005 to 2010, the Moran's I had a decreasing trend, indicating that the spatial agglomeration of the CCD was weakened. From 2010 to 2018, the Moran's I had a slowly increasing trend, indicating that the spatial agglomeration of CCD was slowly increasing.

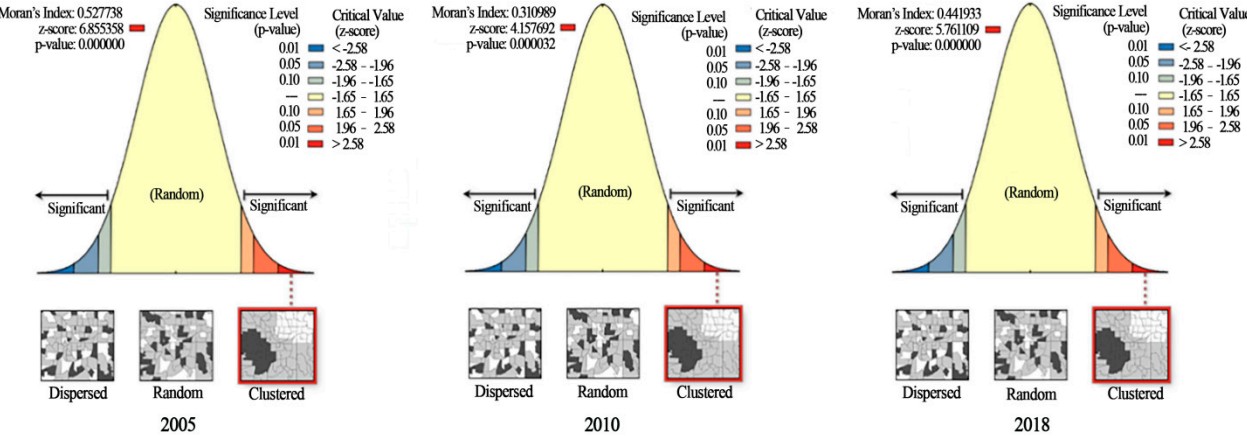

**Figure 7.** Spatial agglomeration of the CCD of CLUB from 2005 to 2018.

In addition, local Moran's I was used to reveal the spatial association type of CCD, and was visualized in ArcGIS 10.6 (Figure 8). In 2005, 2010, and 2018, the spatial distribution of the high–high cluster was basically the same, mainly in the Xining urban area and some surrounding counties. Low–low clusters were mainly distributed in Longxi, Weiyuan, Tongren, and Xunhua. The spatial distribution of the two categories of spatial outliers that were statistically significant (high–low and low–high outliers) was fragmented.

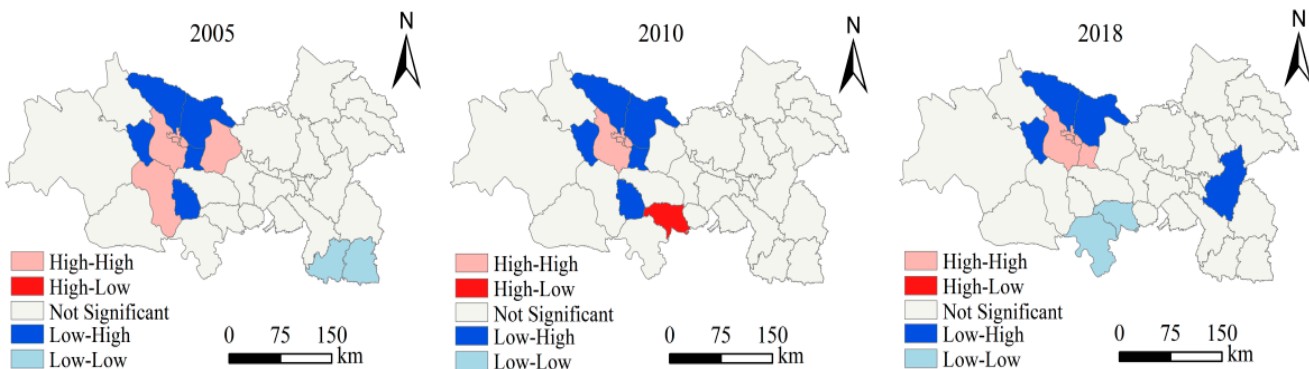

**Figure 8.** Spatial correlation type diagram of CCD of CLUB in the LXUA.

## 5. Discussion

### 5.1. Reasons for the Change of CLUB and CCD Relationship

The evaluation of CLUB is conducive to promoting the intensive and efficient use of land resources and the sustainable development of the region, especially for the northwest region where the ecological environment is relatively fragile. As an important ecological barrier support area in western China, LXUA plays a key role in promoting the high-quality development of the Yellow River Basin, promoting the development of the western region, and ensuring the ecological security of Northwest China [51,52]. The research results show that the CLUB of LXUA is generally low, the polarization of difference is obvious, and the regional development imbalance is a prominent problem. This is consistent with the findings of Shi et al. [53]. Especially, there are significant regional differences in the social and economic benefits. The lack of long-term spatial control measures made Lanzhou and Xining the dominant places. The excessive agglomeration of low-level urban functions, and the limited radiation effect of polarized regions restrict the urban agglomerations integration as well as balanced economic development. In addition, the CCD among the social, economic, and eco-environmental benefit of construction land use increased from 0.48 to 0.55, and is still at a medium and low level coordination stage, which is also consistent with previous research [54–56]. However, the northwest region is currently in a stage of accelerated urbanization. The rapid population growth in some areas has increased the development of land resources. At the same time, the economic structure of the counties is similar, the industrial level is relatively low, and there are still many traditional industries with high input, consumption, and emissions. Coupled with the fragile ecological background and increasing resources, environmental pressure caused most counties to be assessed as lower level regions. This unsustainable development mode needs to be alleviated urgently.

### 5.2. Suggestions for Promoting the CCD of CLUB

With the promotion of ecological civilization construction and the transformation of new urbanization development, combined with the research results of this paper, it has become an inevitable choice for the future land use of LXUA to promote the coordinated development of the social, economic, and eco-environmental benefits. In this process, different types of regions should form their own targeted development paths. For lower-level coordination areas, they should first transform and upgrade the industrial structure to achieve the mutual assistance of urban functions, the industrial dislocation layout, and industrial chain linkage close development, to enhance the economic benefit of land and improve its social and eco-environmental benefits, promote the harmonious and orderly development of construction land in society, economy, and ecology. For medium coordinated regions they should rely on local characteristic industries and regional advantages to first achieve healthy economic development, thereby increasing investment in the construction of basic, urban public services and other facilities, and to promote the improvement of the social benefit of construction land use, so as to improve the overall CCD of such areas. For higher coordination regions, they should strengthen the construction of resource-based industries to continue industrial transformation, and promote the orderly development of the cities, and the development of green, efficient, clean production, and a circular economy. The optimal coordination areas such as Lanzhou and Xining should take the opportunity of the new round of western revitalization to strengthen the intensive use of construction land and improve the infrastructure construction level to accelerate the transformation and upgrading of traditional advantageous industries, to increase the proportion of the tertiary industry, and to consolidate the industrial foundation and employment support for urban development, so that the urban population capacity can be improved, and realize the transformation of the multi-functional benefits of construction land utilization a at a more highly coordinated level.

### 6. Conclusions

This paper uses panel data of 39 county-level cities of the LXUA in China from 2005 to 2018 to construct a three-dimensional system of CLUB, covering society, economy, and eco- environment. We analyzed the temporal and spatial evolution characteristics of CLDI and CLUB and explored the coupling coordination relationship among the benefits. The main conclusions are as follows: (1) The CLDI generally presents a dual-core spatial distribution with Lanzhou and Xining urban areas as the core. In the center of the city, the CLDI is greater. In contrast, in the edge cities, as the distance increases, the CLDI gradually decreases. (2) The social benefit of construction land use showed a "convex"-shaped spatial distribution pattern of "high in the middle and low in the east and west wings". The economic benefit was basically the same as the social benefit in terms of spatial distribution, with great regional differences. The eco-environmental benefit was relatively high andregional differences were small. (3) The comprehensive benefit construction land use had significant spatial differences, and generally presented a gradient spatial structure of "core cities > node cities > general county towns", and the comprehensive benefit was mainly lower level and low level. (4) From 2005 to 2018, the interaction between the social, economic, and eco-environmental benefits of construction land use developed in a coordinated direction in terms of time evolution and spatial correlation, but generally the CCD was still at a medium-low level, and the spatial clustering features were significant.

This paper conducts a more comprehensive analysis of the spatiotemporal evolution of CLDI, CLUB, and the coupling coordination relationship of the multi-functional use benefits of construction land in LXUA, which is of great significance to the sustainable development of future land use of different types. However, there are still some deficiencies that need to be pointed out: (1) Based on the availability of county data, the selection of indicator systems still can be improved. (2) With the update of technical means such as big data, the "flow" element is becoming more and more important. However, this paper does not consider the impact of "data flow" and "feature flow" on regions. (3) Compared with other studies [57–59], this study lacks consideration of more refined data and techniques, and impact mechanisms. In the future, on the basis of comparing and learning from the more mature urban agglomeration paths, a more in-depth analysis and research should be carried out in combination with the characteristics of the study area and the shortcomings of this paper.

**Author Contributions:** Conceptualization, W.Z. and P.S.; methodology, W.Z.; software, W.Z.; validation, W.Z., P.S. and H.T.; formal analysis, W.Z.; investigation, W.Z.; resources, P.S.; data curation, W.Z.; writing—original draft preparation, W.Z.; writing—review and editing, W.Z. and H.T.; visualization, W.Z.; supervision, P.S.; project administration, P.S. and H.T.; funding acquisition, P.S. All authors have read and agreed to the published version of the manuscript.

**Funding:** This research was funded by the National Natural Science Foundation of China (Grant No. 41771130, 42161043).

**Institutional Review Board Statement:** Not applicable.

**Informed Consent Statement:** Not applicable.

**Data Availability Statement:** The data presented in this study are available on request from the first author.

**Acknowledgments:** We sincerely thank the reviewers for their helpful comments and suggestions about our manuscript.

**Conflicts of Interest:** The authors declare no conflict of interest.

## Abbreviations

| | |
|---|---|
| LXUA | Lanzhou-Xining urban agglomerations |
| CLDI | Construction land development intensity |
| CLUB | Construction land use benefit |
| CD | Coupling degree |
| CCD | Coupling coordination degree. |

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
