# Peer review of "Research on Construction Land Use Benefit and the Coupling Coordination Relationship Based on a Three-Dimensional Frame Model—A Case Study in the Lanzhou-Xining Urban Agglomeration"

_land, doi:10.3390/land11040460_

Round 1

Reviewer 1 Report

This paper is presented as an interesting and current study, as the subject of urban/rural land use is approached more and more often, especially in the context of population growth in urban areas and implicitly of urban areas, trying to find a balance between the economy, the environment and the needs of the population.

I appreciate the approach of the subject, which combines statistical methods with methods of spatial representation, which facilitates the understanding of the discussed phenomenon.

However, I believe that for a better understanding of this work some modifications are necessary, which I will highlight below. My analysis was done by chapters, but my remarks will only focus on part of them.

Introduction: This chapter respects the scientific character of such a paper, the general topic of the article and of the study area being well presented, but from this manuscript I cannot tell if the citations are correct or not. The “CLUB” subject has been treated in many researches, so please tell us why did you choose “CLUB” and backup your response through citations from other papers.

Methods: This chapter is a complex one, but the multitude of formulas (around 10) makes the methodology part more difficult to follow.

Also, there are many abbreviations, so please try to include them in a table.

Is there any possibility where you can include an workflow chart? It would help in understanding the process better.

Results: Try to  make your maps bigger, especially figure 3. Also do that with figure 7, it is way too small to understand something from it.

Compare your results with other studies.

Try to resolve the problem with you citations. You have 40 bibliographic  items and  you mentioned in text only 2.

In conclusion, the study is a well-developed one and from my own perspective minor changes should be made only to Introduction, Methods and Results chapters, as I wrote above.

Author Response

Dear expert,

  Greetings!

Thank you for your comments, we have responded to your comments point by point. For details, see the word file below, please review it by experts.

  Good luck in your work and happy life!

  Best Regards,

  Weiping Zhang

Reviewer 2 Report

Excellent job and admirable research were accomplished for the paper. The strong point of this paper is CLUB level and CCD measurement which will help in regional planning for LXUA.

Specific comments:

In line 155 where there was a repetition. 

Author Response

(The authors gave the same response as above.)

Reviewer 3 Report

Dear Authors,

Thank you for the opportunity to review this article.

The title of the study "Research on Construction Land Use Benefit Evaluation and  Coupling Coordination Relationship Based on Three-dimensional Frame Model—A Case Study in the Lanzhou-Xining Urban Agglomeration" corresponds to its content, but the article, should be more professional:

  1. The authors of the article write: "Therefore, based on expounding the coupling coordination connotation and interaction mechanism of construction land use benefits (CLUB), we measured the CLUB level and the coupling coordination degree (CCD) between its principal components in LXUA from 2005 to 2018" We currently have 2022. Therefore, please extend scientific research by 2019, 2020 and 2021.

  1. In the Introduction chapter, no citation of articles from the bibliography. This cann’t be, it is unacceptable. No article provided in the bibliography does not find a place in the article text (total lack of quotations). This must be improved.

  1. This publication contains mostly research which are analyzed in the China. However, the introduction lacks a reference/ comparison between China and other countries such as the United States of America or Europe. The same goes for references.  References include 40 publications are cited in the entire article. In my opinion, this is not enough.  It is proposed to extend the bibliography in this regard to the publications of authors from universities from other European and USA. Please complete this and the article will be a valuable scientific contribution.

  1. Similarly, the discussion or conclusion should refer to research conducted in this field in other countries. Must complete this and the article will be a valuable scientific contribution.

  1. Figure 7. Not readable. Please correct it.

In relation to the above, I express the opinion that the work submitted for review should be published after taking into account the comments of the reviewer. Reconsider after major revision.

Author Response

(The authors gave the same response as above.)
